# Integration and Field Evaluation of an IoV System for Enhancing Road Safety

**Aristotelis Spiliotis [1] , Fotios Giannopoulos [2], Christos Spandonidis [2,\*] , Maria Gkemou [1] and Natalia Kalfa [3]**

1   Center for Research and Technology Hellas, 57001 Thessaloniki, Greece
2   Prisma Electronics S.A., 68100 Alexandroupolis, Greece
3   Attikes Diadromes S.A., 19002 Paiania, Greece
\*   Correspondence: rdprojects@prismael.com

**Abstract:** Road safety is a major global concern, as millions of lives are lost every year because of road accidents. Towards an effort to increase road safety, several Internet-of-Vehicle systems have been developed over the last years in order to better monitor vehicle and driver behavior and issue warnings that effectively prevent life-threatening accidents. These systems face a number of challenges including connectivity issues and high installation and/or maintenance costs. The current work introduces the ODOS2020 system, an integrated Internet-of-Vehicles system aiming to increase road safety. The system comprises several On-the-Road Units for vehicle-related data collection from affordable, energy-efficient magnetometers and calculation of critical parameters, such as each passing vehicle's speed and direction. A Road-Side Unit accumulates data from the On-the-Road Units, sends data to a cloud infrastructure for further analysis and sends dedicated warnings to the drivers based on their road behavior and/or specific traffic conditions via a dedicated Human–Machine Interface. The overall system architecture and the key features of its modules are being presented, as well as the evaluation results of specially designed tests performed in an actual motorway under real use case scenarios. The evaluation results showed both a very good technical performance of the system and a high level of user acceptance. This in turn means that the system can be employed for effective traffic control and road accident avoidance via monitoring of critical vehicle parameters and early warning of the drivers based on their and other drivers' behavior, road conditions and real-time, unpredictable events.

**Keywords:** internet of vehicles; traffic monitoring; traffic management; road safety; road accident prevention

## 1. Introduction

Road safety is a primary global concern, as the World Health Organization reports that approximately 1.3 million people die each year as a result of road traffic crashes [1]. For the European Union (EU) alone, the number of vehicle accidents that resulted in death before 2020 was constantly over 20,000, with the European Commission establishing its 'Vision Zero' plan, an attempt to virtually eliminate traffic fatalities and serious injuries by 2050. Towards materializing its long-term goal, the EU established road safety guidelines dating back to 2010, aiming to reduce European road deaths by 50% by 2020 [2]. Even though significant progress has been made towards this goal with the total number of road traffic-related fatalities constantly decreasing since 2010, in 2017 the EU stated that reaching the objective of zero road fatalities by 2050 will be very challenging. The EU also acknowledged that the persistently high number of road traffic fatalities and serious injuries is a major societal problem, causing human suffering and unacceptable economic costs [3]. The number of people who died in road accidents in the EU in 2020 was 18,786, of which 44% were passenger car occupants and 16% were motorcycle occupants. While this constitutes a decrease in the number of people killed in road traffic accidents in the EU

by 17% compared with 2019, this drop is largely due to restrictive measures on passenger transport due to the impact of COVID-19 [4]. Both the World Health Organization and the EU identify the lack of efficient speed control as one of the most important causes of road traffic accidents [1,5].

Towards improving road safety, Internet of Vehicles (IoV) provides various technological solutions, employing a wide range of sensors, including LIDAR and ultrasonic sensors for collision avoidance [6], video or image-capturing sensors [7–9], radar sensors [10], inductive loops, and magnetic sensors [11]. Sensors data combined with communication systems for effective communication between vehicles (Vehicle-to-Vehicle or V2V communication), as well as between vehicles and infrastructure (Vehicle-to-Infrastructure or V2I communication) create effective IoV solutions for road safety [12,13]. Due to the nature of each situation that each IoV system addresses, different communication systems and protocols can be employed, to better serve the target application [14]. In addition to novel sensors networks and communication systems, data analytics and data processing algorithms, often running on the edge of IoV networks, are being employed for effective and quick systems response [7] and/or early warning of the end users, often based on data related to the drivers' behavior, prior violation and accident records [15].

Cooperative Intelligent Transport Systems (C-ITS), is a category of Intelligent Transport Systems, made possible due to IoV. They enable communication between sensor systems, existing infrastructure and a variety of end users, ranging from the drivers themselves to traffic management center (TMC) operators, providing increased road safety and in-time detection of road infrastructure critical deficiencies [16]. However, at times, their integration with the existing infrastructure and/or their installation is associated with a rather high cost, ultimately undermining their use [17]. Because of that, numerous recent pieces of research focus on low-cost IoV systems, often employing roadside units with various types of MEMS sensors that are both affordable to produce in large quantities and energy efficient, making the overall solution more cost efficient [18,19]. Among these systems, those employing magnetometers seem to be very popular, with a comprehensive review provided in [20]. Other systems employ artificial intelligence and power autonomous systems in order to effectively adjust the processing power on the edge and distribute it among various units to achieve power efficiency and lower operational cost [21]. Another important aspect of C-ITS is communication, with the recent focus being in the development of Vehicle Ad-hoc Networks (VANET), an adaptation of Mobile Adhoc Networks (MANET) for ITS. Several challenges are associated with VANET, such as pattern mobility, reply RSU and high movement speed among others, while various approaches have been developed in order to enhance VANET's ability and capacity, which is well documented in [22].

The ODOS2020 project introduces an integrated IoV system comprising several On-the-Road Units (ORUs) for collecting data from sensors embedded seamlessly on the road, calculating various passing vehicle movement parameters, such as their movement direction and speed, while sending information to a Road-Side Unit (RSB) for further analysis. The end users are informed via a dedicated Human–Machine Interface (HMI) that communicates with the RSB warns the end user using visible and audible stimuli. The main novelty of the developed system lies in its versatility in monitoring various types of vehicles, such as cars and motorcycles and in effectively warning their drivers regarding potentially dangerous road behavior, despite the type, age and communication systems of their vehicle. At the same time, it is scalable, as the system's core architecture can be expanded straightforwardly, and is easy to install. The present work focuses on the ODOS2020 system integration and evaluation in real use case scenarios. The overall system architecture and the characteristics of each one of its modules are presented in Section 2. The Human–Machine Interface used for communication with the end users is presented in Section 3. Dedicated pilot tests were performed in one of the biggest motorways in Athens, Greece, testing the entire system's performance on various real use case scenarios. The field evaluation plan, alongside the corresponding results, including technical performance and user acceptance results, is presented in Sections 4 and 5, respectively. Section 6 includes the

discussion regarding the test results and presents challenges and the authors' future plans regarding further expansion and improvement of the developed system, while Section 7 summarizes the most important points of the presented work.

## 2. System Architecture

### 2.1. General Architecture

The ODOS2020 integrated system comprises several modules and is depicted in Figure 1.

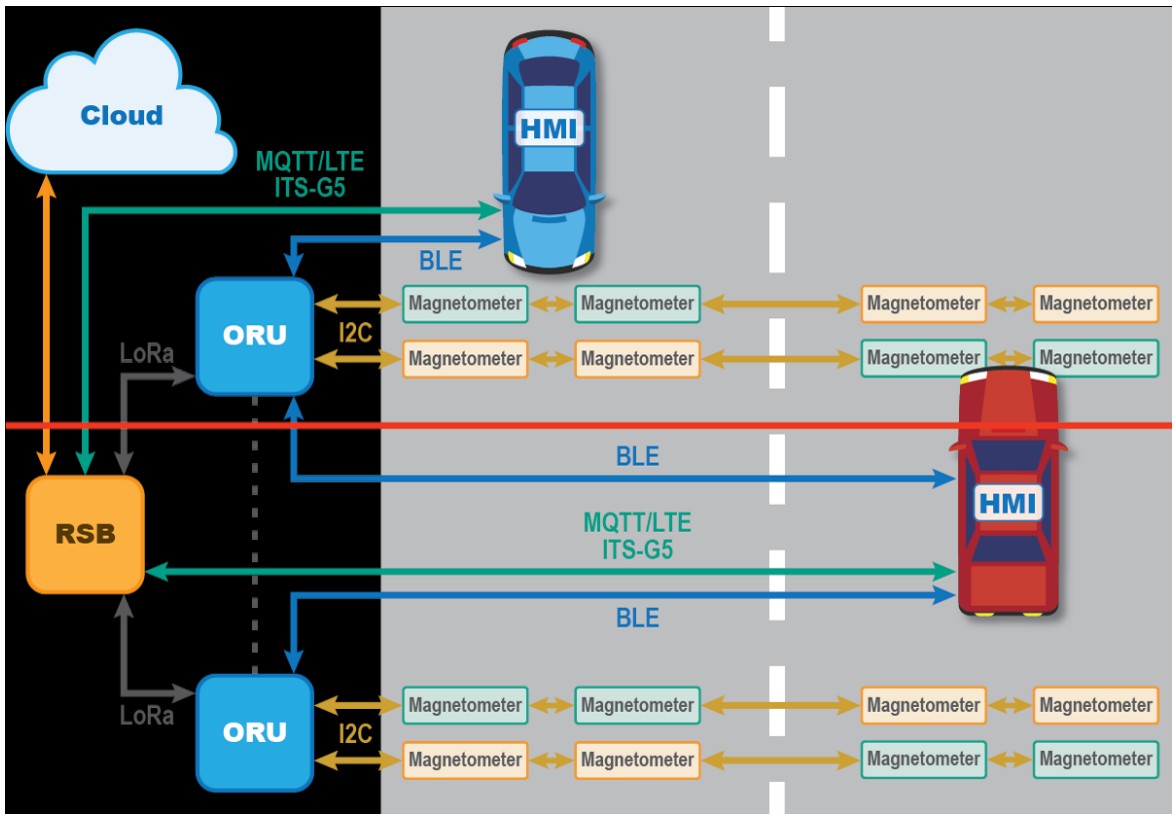

**Figure 1.** ODOS2020 system architecture.

Several pairs of sensor stripes comprising MEMS magnetometers detect passing vehicles and help to determine several critical parameters of their movement, such as their speed and direction. Sensor data are collected by On-the-Road Units (ORUs) using the versatile and highly scalable I$^2$C protocol. The ORUs calculate the critical parameters of the vehicles' movement and send them to a Road-Side Unit (RSB) via LoRa. The ORUs also communicate with the passing vehicles via Bluetooth Low Energy (BLE) to identify passing vehicles so that personalized information can then be sent to each specific vehicle from the RSB.

Each RSB acts as a central point where data from several ORUs are collected and from there are sent to the Cloud for further processing. The RSB also communicates with each passing vehicle via MQTT over LTE for vehicles with older communication capabilities (non-equipped vehicles) and via ITS-G5 for vehicles with modern communication systems that allow connectivity via BLE, ITS-G5 and several other protocols (equipped vehicles).

A dedicated HMI is used in each passing vehicle in order to display warnings issued by the system to the driver. The HMI displays useful warnings in a simple yet intuitive way using easy to understand images and sounds so that the driver can avoid a potential accident by accordingly adjusting his/her behavior. The HMI has been developed so that it can be used with both contemporary and older vehicles. It can be installed in the existing hardware infrastructure of the vehicle or the driver's mobile device.

Several aspects of the developed system's scalability and versatility can already be traced in this initial description of the system's general architecture. The use of the $I^2C$ protocol for communication between the ORUs and the sensors located on the road allows for easy expansion of the system's sensing capabilities in several additional traffic lanes, without the need for additional cabling, as the $I^2C$ protocol employs only two (2) wires to form a bus where several sensors can be connected. Moreover, several ORUs can be placed along a road or highway/motorway and be spaced according to the traffic conditions of the road in order to effectively monitor the road's traffic. The LoRa protocol employed for communication between the ORUs and the RSB can be used to cover a wide area and to support a large number of connected devices, as long as the data exchange between them remains within certain limits dictated by the protocol limitations [23]. In this way, data can be accumulated for processing in a centralized cloud infrastructure with very few communication units deployed. The versatility of the developed HMI makes the system easy to integrate into practically every vehicle and easily used by nearly every driver.

In the following sections, each module of the ODOS2020 system is presented in greater detail, outlining the special features of each and its specific role in the entire system.

### 2.2. On-the-Road Unit (ORU)

The architecture of each ORU is presented in Figure 1. The magnetometers were embedded in very small printed circuit boards (PCBs) especially developed for the project, including low-profile electronic components and durable cable connectors that can be encapsulated in a protective material, installed on a road and withstand everyday use in such a harsh environment. The characteristics of the employed sensors, Melexis' MLX90393, are presented in Table 1 [24]. As can be seen from the table, magnetometers of rather low power consumption, profile and cost were selected so that they can be replaced with a small cost if needed.

**Table 1.** Technical characteristics of the MLX90393 magnetometers.

| Parameter | Value |
|---|---|
| Digital resolution | 16 bits |
| Dynamic range | 5–50 mT |
| Maximum sampling rate | 500 samples per second |
| Maximum current consumption (xy-axis acquisition) | 3 mA |
| Maximum current consumption (idle mode) | 5 μA |
| Packaging | QFN 3 mm × 3 mm |
| Average unit purchase cost (single unit/5000 units) | 3 €/1.30 € |

Each ORU is connected to two (2) sensor strips placed vertically to the traffic flow of the road. Therefore, for two traffic lanes, as shown in Figure 1, every ORU collects data from eight (8) magnetoresistive sensors, four (4) for each lane, using the $I^2C$ communication protocol. A total of three (3) traffic lanes can be covered with the existing design, due to the $I^2C$ protocol's bus length limitations. Two of the four sensors of each lane, depicted in green in Figure 1, detect the passing vehicles first, while the other two, depicted in orange in Figure 1, detect each passing vehicle last. From the sequence in which these groups of sensors detect each passing vehicle and by measuring the time difference between two consecutive sensors excitation, critical parameters regarding the driver's behavior, such as the vehicle's speed and its direction can be calculated, given that the sensors strips are placed on the road with a fixed distance between them. In order to be detected by the ORU, the vehicle must be moving at a speed greater than 15 km/h. Four sensors are used per lane to cover every vehicle moving within the margins of the lane. The system can detect several vehicles at the same time.

The developed ORU can communicate wirelessly via BLE and LoRa. The BLE is used for uniquely identifying each passing vehicle. This helps the system to collect vehicle-specific data and in turn provide personalized warnings to each driver, based on his/her

behavior. Each ORU is also equipped with a LoRa communication module for sending data to the RSB.

The developed ORU was based on Prisma Electronics' PrismaSense™ universal data acquisition platform [25], adapted for data acquisition from MEMS magnetometers and communication via BLE and LoRa. More on the ORU, its features and the algorithms running on it are provided in [26].

### 2.3. Roadside Unit to Vehicle Communication Unit

Depending on whether the passing vehicle is equipped or non-equipped, the information sent used the corresponding communication channel. Equipped vehicles communicate with the mediator through the ITS-G5 network, while non-equipped vehicles—through the MQTT Broker and via the LTE network. To meet the communication demands, the mediator is equipped with both ITS-G5 and LTE communication modules.

For the C-ITS solution of the equipped vehicles, a transceiver for direct short-range communication (DSRC) with the mediator is necessary to establish direct communication with the passing vehicle. For this purpose, the THEO-P1 by u-blox was selected based on its technical specifications (see Table 2), which cover the requirements for V2X communication. From the side of the roadside unit, a Cohda MK5 is used for ITS-G5 communication.

**Table 2.** Technical characteristics of the DSRC module.

| Name | Type | Operating Frequency | Maximum Output Power | Maximum Data Transfer Speed | Maximum Power Consumption |
|------|------|---------------------|----------------------|-----------------------------|---------------------------|
| THEO-P1 series | V2X transceiver | 5.9 GHz (5.85–5.925 GHz) | +23 dBm | 27 Mbps | 4 Watt |

The information on environmental conditions and road events is contained in DENM messages, transmitted to the equipped vehicles through the ITS-G5 protocol [27] along with a CAM message [28], sent every 100 ms, to inform the passing vehicle of its presence. In return, the mediator informs the passing vehicles about the topology of the road, every second, with a MAP message. MAP messages comprise a common ITS PDU header and mapem [29] data.

For the non-equipped passing vehicles, the mediator has to communicate with the MQTT broker, which in turn, communicates with the application (see Section 3.4) installed on the terminal device of the vehicle. This communication between the MQTT broker and the mediator is established through the LTE network. To this end, a u-blox TOBY-L210 module is chosen, since the technical characteristics comply fully with the requirements set.

## 3. Human–Machine Interface Architecture

In recent years, new and promising approaches exist to improve the interaction between drivers and vehicles. Among them, new approaches exist in the user interfaces of smart devices. The availability of fast and reliable wireless communication systems (e.g., Bluetooth, 4G and 5G) has enabled the development of driver interaction systems that reside in smart devices in addition to traditional vehicle equipment. Many examples include the use of smartphones as additional screens for driving-related information [30].

Critical information for road safety is provided traditionally through in-vehicle messages in the form of warnings, which however raise ergonomic issues regarding the driver's workload, distraction [31], understanding or combination of warnings (of varying severity) provided by coexisting applications.

### 3.1. Design Framework

The HMI is designed to function as part of a C-ITS that targets both equipped and non-equipped vehicles seamlessly for all types of vehicles and various applications, which may comply with different traffic scenarios. To fulfill these requirements, five basic principles were determined for the interface that must be satisfied:

(a)    Support a distributed architecture for a C-ITS;
(b)    Support a multi-level strategy for in-vehicle information system;
(c)    Provide the required information without creating or distracting the driver;
(d)    Support personalized/targeted information.

The architecture of the modules that make up the HMI reflects the distributed architecture of the integrated system. The architecture incorporates a standard messaging protocol based on the "publish" and "subscribe" functions (Message Queuing Telemetry Transport—MQTT). This protocol is designed for remote connections of devices (e.g., sensors) with limited resources and limited bandwidth.

As seen in Figure 2 the HMI is divided into two logical elements: one materializes the high-level architecture and the other handles the co-existing applications. All elements are connected to the MQTT broker, which is the central node for all communications. Last, the HMI is being implemented on mobile applications for smart devices (Android and iOS) and through a Decision Support System (DSS) which takes over the prioritization of the messages/warnings, when these co-exist, and chooses which will be sent and when to each device.

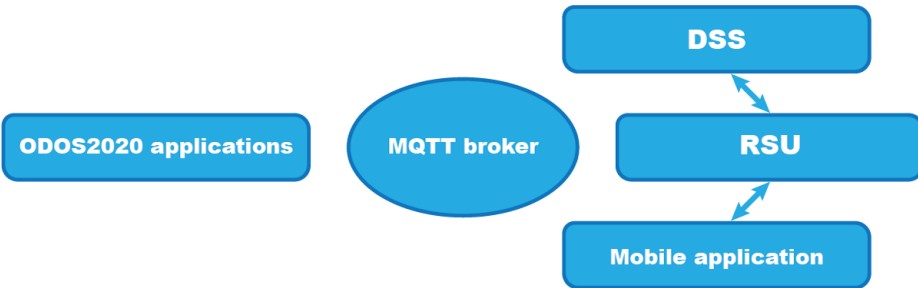

**Figure 2.** HMI system architecture rationale.

### 3.2. Warnings and Message Flow Architecture

The HMI is designed to function as part of a C-ITS which targets both equipped and non-equipped and must have seamless communication with different discrete devices at the same time. To achieve this, it aims at taking advantage of the vehicle-infrastructure sensing and communication components and thus optimizing the security and effectiveness of the provided information through a decentralized decision-making architecture. Consequently, the dynamics of the user interface rely more on fast connectivity and less on the computing power of the devices themselves.

Figure 3 illustrates the flow of warnings/messages for the scenario of road works taking place in the lane of a passing vehicle.

### 3.3. User Interface Design and Standardization

In order to cover the basic design principles of the system, the user interface design was made by dividing the screen into two parts. As part of the user interface design approach, each "screen" consists of two parts, upper and lower: the upper part of the user interface explains "Why" the driver should react, while on the lower part of the user interface an explanation of "What" the driver should do is displayed.

Each section (top and bottom) of the basic approach is divided into individual sections as shown in Figure 4.

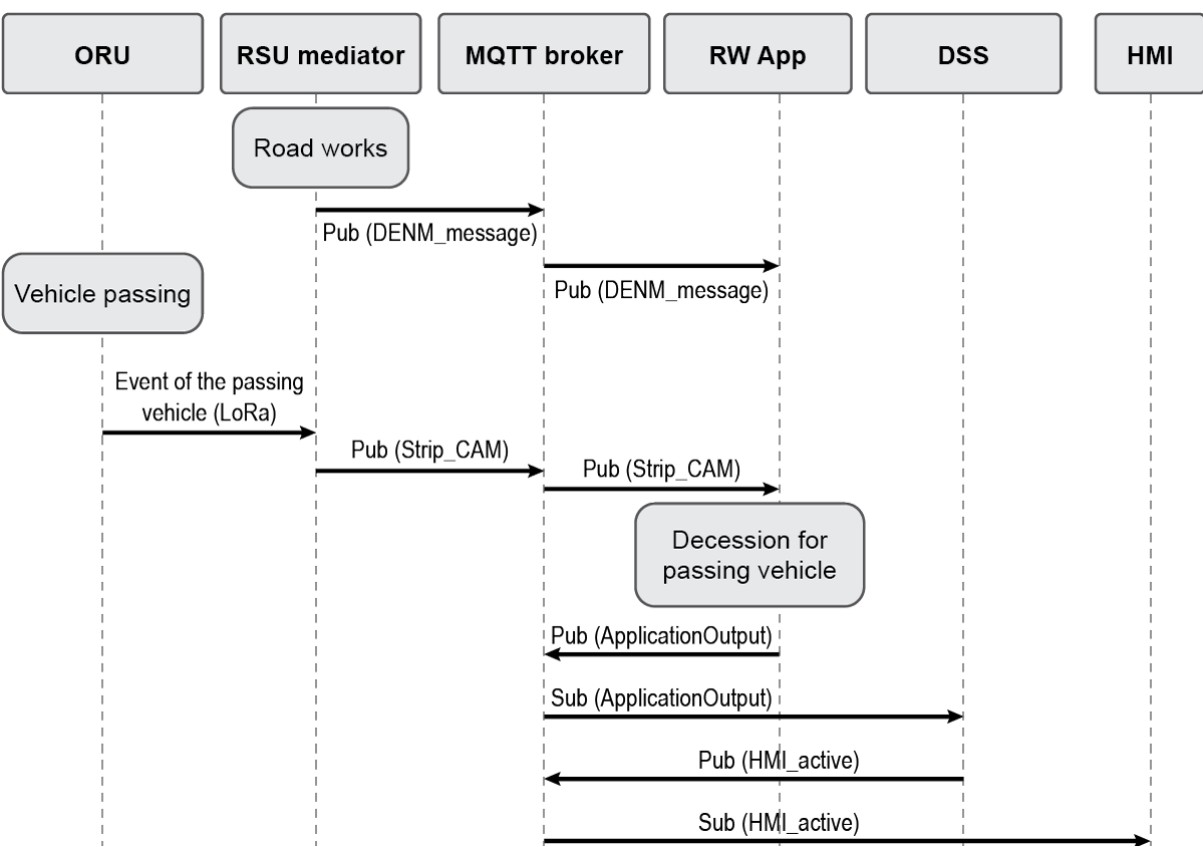

**Figure 3.** Warnings & messages flow architecture example.

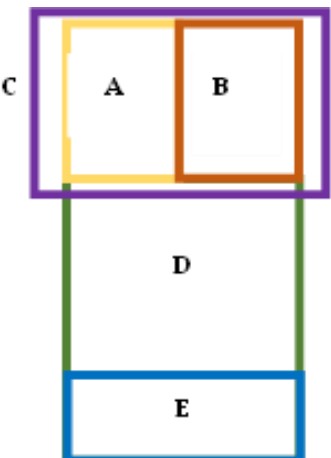

**Figure 4.** Display the principle of the HMI.

where,

$$A = text = [widget\ 1\ |\ widget\ 2|\ \ldots\ |\ widget\ N], \tag{1}$$

This text verbally informs the driver of the upcoming danger.

$$B = widget = [text\ 1|\ text\ 2|\ \ldots\ |\ text\ N], \tag{2}$$

This area informs the driver with an image of the upcoming danger.

$$C = color = [blue\ |\ orange\ |\ red], \tag{3}$$

The overall color of the screen is different according to the level of danger.

$$\mathbf{D} = \text{dynamic display} = [\text{level 1} \mid \text{level 2} \mid \ldots \mid \text{level N}], \tag{4}$$

This area presents to the driver the reaction proposed by the system (recommended driving maneuver).

$$\mathbf{E} = \text{text} = [\text{text 1} \mid \text{text 2} \mid \ldots \mid \text{text N}], \tag{5}$$

This part verbally informs the driver of the reaction proposed by the system (driving maneuver recommendation).

### 3.4. Mobile Applications

The mobile applications (Android and iOS) are implemented as multi-threaded applications. They are used as the terminal devices of non-equipped vehicles and each contains three main components. The first component, called Location Tracking Service (LMS), is a service that runs in the background and implements communication with the device's GPS hardware. The element operates continuously and receives every second the longitude, latitude, altitude, direction and speed. After that, the data is provided in the third element.

The second component, called MQTT Service, is a service that runs in the background and implements the communication with the MQTT broker in a separate thread. The service subscribes to the topic "ODOS2020/HMI_active/#{StationID}" to receive the messages provided by the DSS. Messages are encoded and decoded using the C programming language as a structure. Thus, the mobile application implements a specific wrapper/module that can be used by all components to post and receive messages. The messages received by the mobile application are of type HMI_active (Figure 5). The service then decodes the message using the module and converts it to a string sent to the third component.

```c
typedef struct {
  uint64_t UTC_time;
  uint32_t ApplicationID;
  uint32_t StationID;
  uint8_t  WarningLevel;
  double   TargetDistance;
  double   TargetSpeed;
  int32_t  TargetClass;
} HMI_active;
```

**Figure 5.** Structure HMI_active.

The third component implements the entire functionality responsible for presenting the User Interface (UI) elements and audio messages to the driver/rider. This component runs on the main thread. The component also contains two broadcast receivers to receive messages from the other two components (Location Tracking Service and MQTT Service). The first broadcast receiver is responsible for receiving the data provided by the LMS and the second for receiving data from the MQTT Service.

## 4. Evaluation Plan and Field-Testing Approach

ODOS2020 adopted two different types of testing: (a) The integration tests that verified the smooth operation of the individual parts, which included only the technical evaluation of the system and took place during the development phase of the system, before the conduction of the pilot tests. (b) The field tests involved the user trials on-site with users/drivers in real conditions. The latter collected combined information to provide a holistic evaluation of the system both technically as well as from the users' acceptance perspective.

*4.1. Methodology and Evaluation Framework of the User Trials*

The described system has been evaluated in the field through user trials carried out in two (2) rounds in a semi-controlled traffic environment based on the FESTA methodology as described in the FESTA handbook [32]. Figure 6 reflects the methodology used for the evaluation of the system.

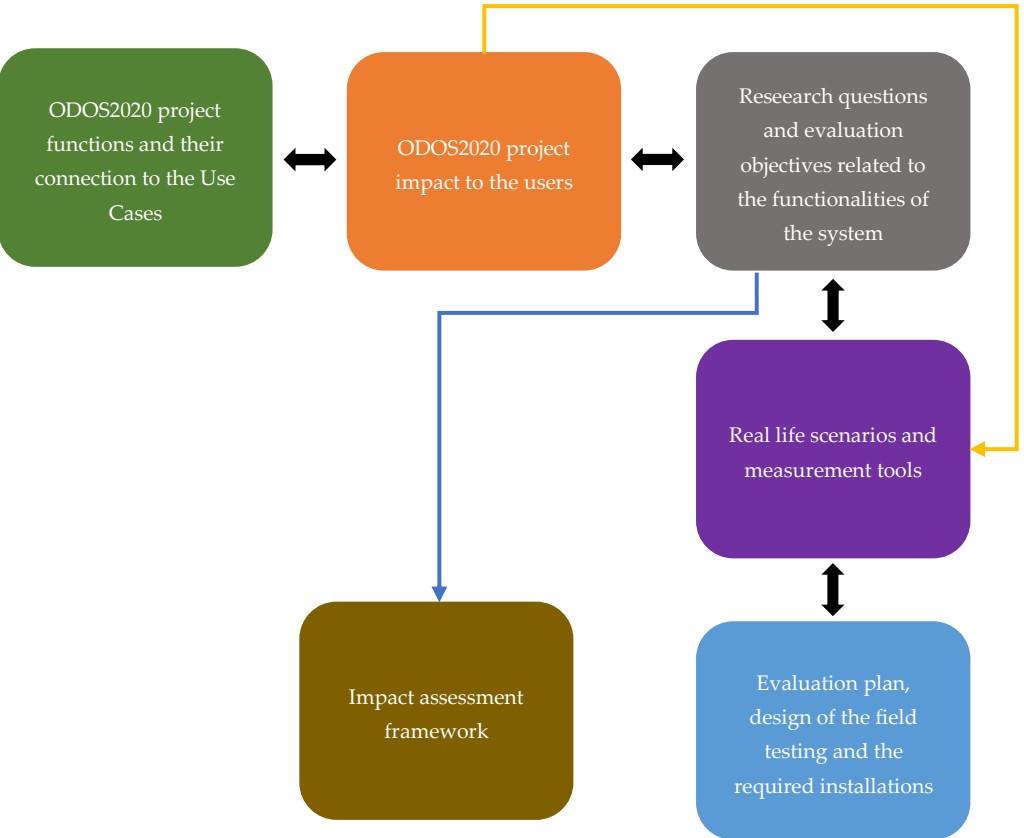

**Figure 6.** The ODOS2020 evaluation framework.

The functions of the system are defined and matched to the use cases of the project [33] and the definition of the research questions that need to be answered was drawn up along with the evaluation objectives. These were linked to the real-life scenarios and necessary measurement tools to conclude the evaluation plan and the required installations. The functionalities of the system, their connection to the project use cases and the defined research questions are not presented in this paper as their description is throughout presented in the respective Deliverable EED1 [34].

To evaluate the overall performance of the system, a two-fold methodology was developed based on (a) objective measurement tools, based on multiple data logging mechanisms incorporated within every component of the system that record every event and (b) subjective evaluation tools consisting of questionnaires addressed to the users before and after the conduction of the tests, which have been selected to evaluate subjectively, but with objective weighted criteria, the user's experience. In each pilot test, a sequence of asynchronous events is created, those describing the conditions and those describing the movement of the vehicles. Each event is converted into the appropriate message and transferred between the subsystems either directly or by using the cloud. Each generated message includes the timestamp of its creation. This creates a chain of events, which is related to one another. The time of creation and reception of these messages is offered as the basic tool for evaluating the performance of the system regarding the timely information of the users about the upcoming situation on the road.

### 4.2. Pilot Site Preparation and Conduction Plan

The user trials were carried out on the Attiki Odos motorway, a modern, three-lane in each direction, closed-tolled motorway of 70 km that connects 28 municipalities of the Attica prefecture. The location chosen was between P10.8 and P11.1 km of the western peripheral part "Ymittos", in the direction of Rafina and just before exit Y8 (Pallini). Figure 7 shows the location, which served as the pilot site in which the installed tapes of the system are marked in blue.

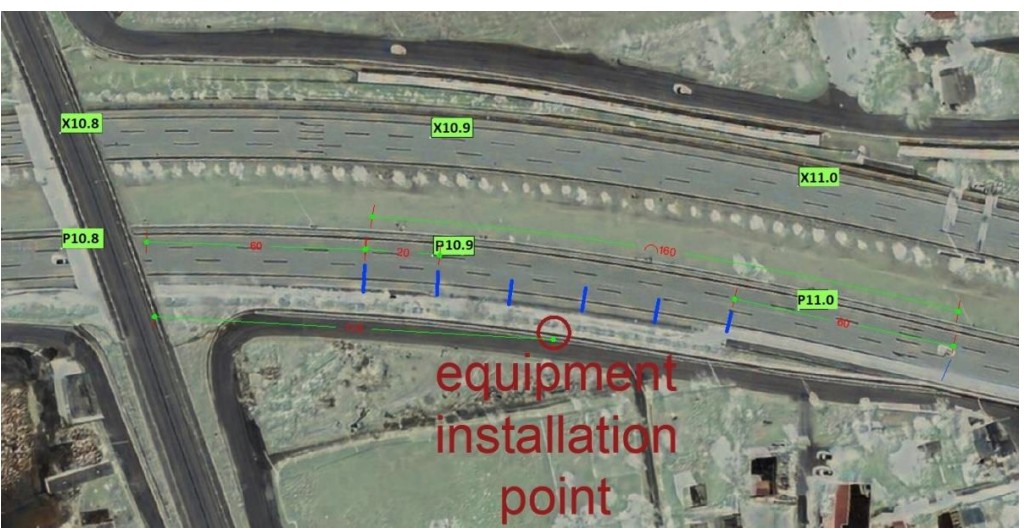

**Figure 7.** The installation site.

The iterative evaluation approach of the user trials was split into three phases, as depicted in Figure 8. The first included the technical verification of the system, which took place after the installation of the system on-site.

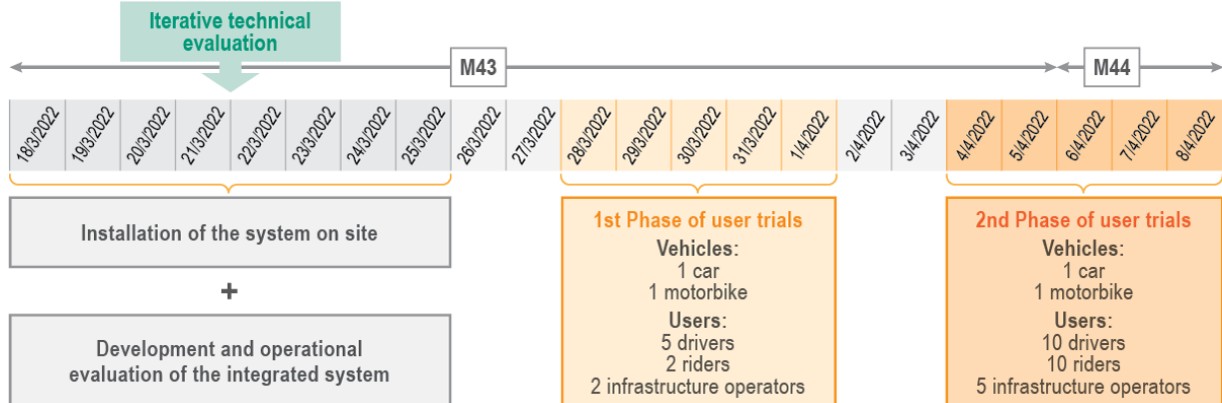

**Figure 8.** Conduction plan of the user trials.

### 4.3. User Trials Plan

The plan was prepared according to the five steps below:

Step 1: Before each evaluation round at the test site, the feasibility of the evaluation scenarios was re-examined as well as the appropriateness of the evaluation protocol together with the mechanisms, which were set to capture the metrics defined per scenario;

Step 2: Before conducting the user trials, the site managers received instructions from the trial coordinator on how to perform each experimental procedure summarized below;

Step 3: Personnel responsible for the ethics had already reviewed the trials for compliance with the project Ethics Policy;

Step 4: Before the official user testing process, full trial runs were performed throughout the system setup with an experienced user-technician to ensure that everything was in place for testing (i.e., pre-test);

Step 5: Users performed the evaluation scenarios. Before the execution, they filled in a consent form and a questionnaire, which would assist in the evaluation of the system from the user-acceptance perspective. A representative from the trials coordinator was present in the vehicle (not applicable in the case of the motorbike) during the execution of the scenarios, while another representative was outside the vehicle to monitor the process. An event log with information about recorded actions was used as a cross-reference for the important information produced within the logs of the system;

Step 6: After completion, the users were asked to complete another questionnaire (anonymously) for further evaluation of the system.

## 5. Results

The results of the pilot tests can be distinguished into two (2) categories. Those concerning the performance of the system, i.e., the extent to which the integrated system can collect and transmit the necessary data to fulfill the scenarios and the results concerning the user acceptance.

### 5.1. Technical Performance Results of the System

This section presents the performance results of the ODOS2020 system during the pilot tests that took place on the Attiki Odos motorway.

#### 5.1.1. Vehicle Detection and Speed Calculation System Performance

The first type of quantitative system performance test performed targeted the success rate of the system in identifying a passing vehicle, its direction and speed, which are its main critical parameters of interest. For this reason, during the tests performed using both a car and a motorcycle, each vehicle passed five (5) times from each ORU position at a certain speed, with the speed varying between 15 km/h and 50 km/h. The system's performance in detecting each vehicle, its direction and speed is summarized in Tables 3 and 4. For the calculation of the speed detection error percentage, only the results of valid direction detections were taken into account.

**Table 3.** Vehicle and direction detection results.

| Car | | | | Motorcycle | | | |
|---|---|---|---|---|---|---|---|
| Speed (km/h) | Passes | Vehicle Detection | Direction Detection | Speed (km/h) | Passes | Vehicle Detection | Direction Detection |
| 16 | 30 | 30 | 23 | 15 | 30 | 30 | 22 |
| 26 | 30 | 30 | 28 | 30 | 30 | 30 | 27 |
| 46 | 30 | 30 | 30 | 50 | 30 | 30 | 29 |

**Table 4.** Vehicle speed detection results.

| Car | | | Motorcycle | | |
|---|---|---|---|---|---|
| Speed (km/h) | Passes | Average Speed Detection Error (%) | Speed (km/h) | Passes | Average Speed Detection Error (%) |
| 16 | 30 | 11.6% | 15 | 30 | 6.4% |
| 26 | 30 | 11.3% | 30 | 30 | 13.1% |
| 46 | 30 | 10.3% | 50 | 30 | 4.8% |

As can be seen from Tables 3 and 4, both types of vehicles were successfully detected by all the ORUs on the field at all times. The vehicle direction detection success rate is

very high as well, while the average speed detection error remains constantly under 14%, demonstrating a cohesive and good system performance. To better evaluate the system's performance regarding vehicle speed estimation, the fact that the speed indication reference during the tests was the corresponding vehicle's speed gauge should also be taken into account. The speed gauge has an inherent deviation between 3 km/h and 5 km/h from the speed it displays.

### 5.1.2. Communication System Performance

The system aims to address, as already mentioned, both equipped and non-equipped vehicles. The equipped vehicles have installed systems that implement V2X communications, while the non-equipped do not have any other type of equipment except a smart mobile phone to receive and display information intended for the end user. Since the used differences between the two approaches are large concerning the used telecommunication technologies, examples of the measured performance of the system are presented separately for each case.

Figure 9 depicts the time between the creation of the awareness message (CAM) (which contains both the latest geographic position and the current motion parameters and feeds the applications) to the HMI notification. The non-equipped vehicle follows a different procedure which is measured based on the Figure 10 time intervals.

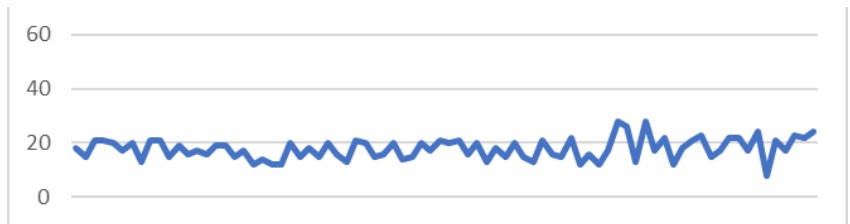

**Figure 9.** Time delay between CAM message and driver notification on the equipped HMI display in ms.

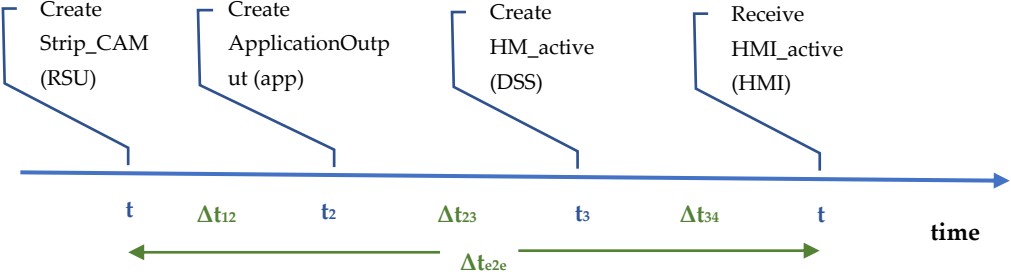

**Figure 10.** Sequence of messages in the system for the non-equipped vehicle.

The following graphs (Figures 11–14) present examples of the results regarding these time frames, as derived from reading taken during the pilot user trials of the system.

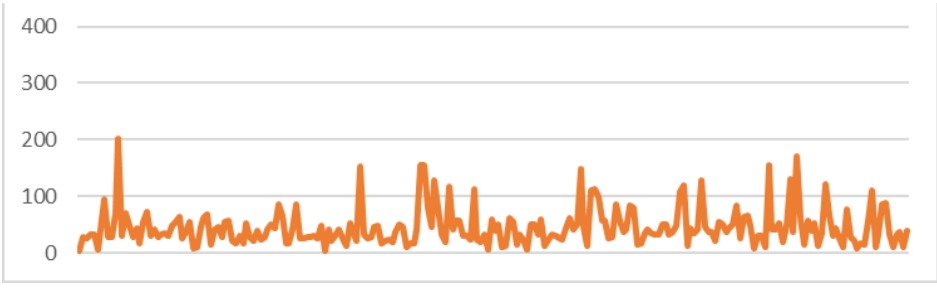

**Figure 11.** Time delay between detection and application exit ($\Delta t12$ vs. ms).

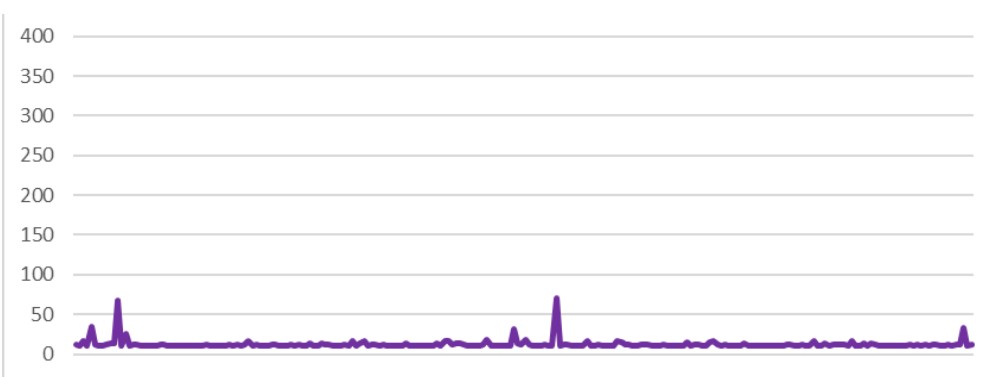

**Figure 12.** Time delay between the application output and the DSS output ($\Delta t_{23}$ vs. ms).

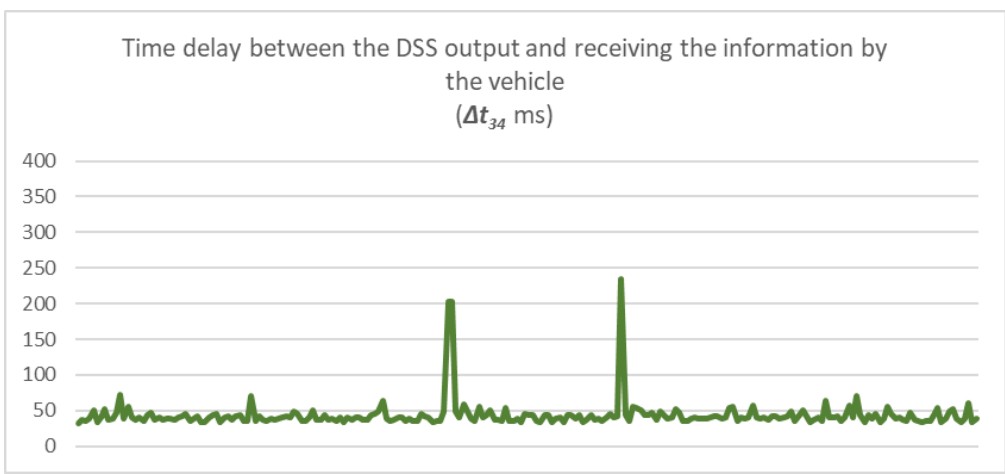

**Figure 13.** Time delay between the DSS output and receiving the information by the vehicle ($\Delta t_{34}$ vs. ms).

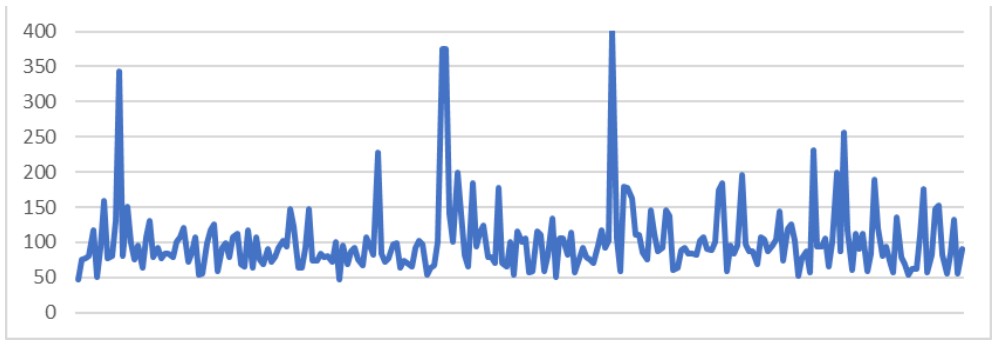

**Figure 14.** Total time delay of the non-equipped vehicle between detection and HMI display ($\Delta t_{e2e}$ vs. ms).

### 5.2. User Acceptance

The first dimension of user acceptance is obtained by the comparison results between the logging data of the system and the event logs taken during the user trials. This measurement consists of three evaluable indicators, the degree of success, compliance and the reaction type. The degree of success is understood as the extent to which the tests were carried out correctly and is determined by whether the system produced all the required data, transmitted those according to the original plan and arrived as information to the user. The degree of compliance refers to the extent to which the user complied with the instructions or information received. Full compliance indicates that the driver

reacted according to the presented instruction; partial compliance indicates that there was a reaction but not necessarily according to the instructions, while no compliance means that the information had no effect on the driving behavior of the user. The following figures (Figures 15–17) show the resulting diagrams for the VMS scenario based on all tests that took place for all types of vehicles. At this point, it should be noted that user acceptance is not differentiated based on whether the car vehicles are equipped or non-equipped, as in any case, the HMI perceived by the user does not differ.

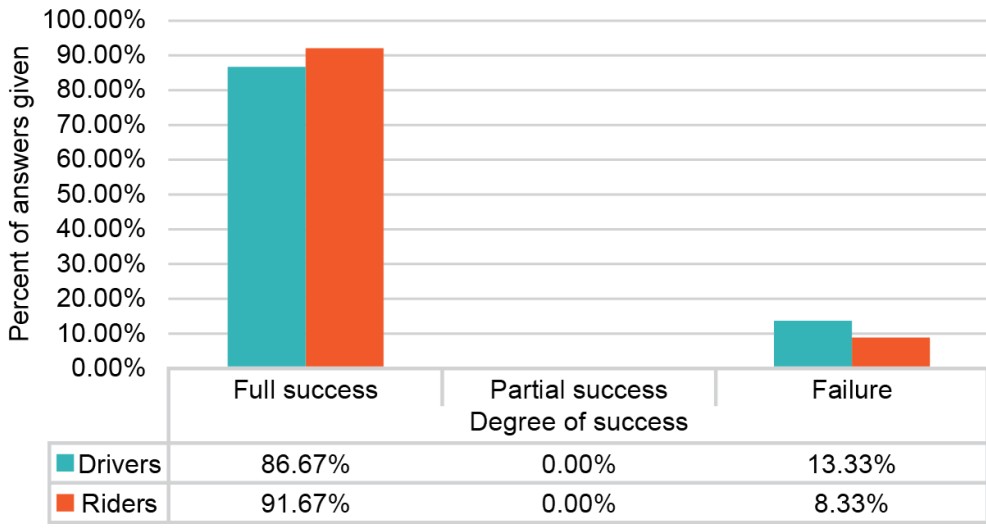

**Figure 15.** Degree of success for the scenario "VMS".

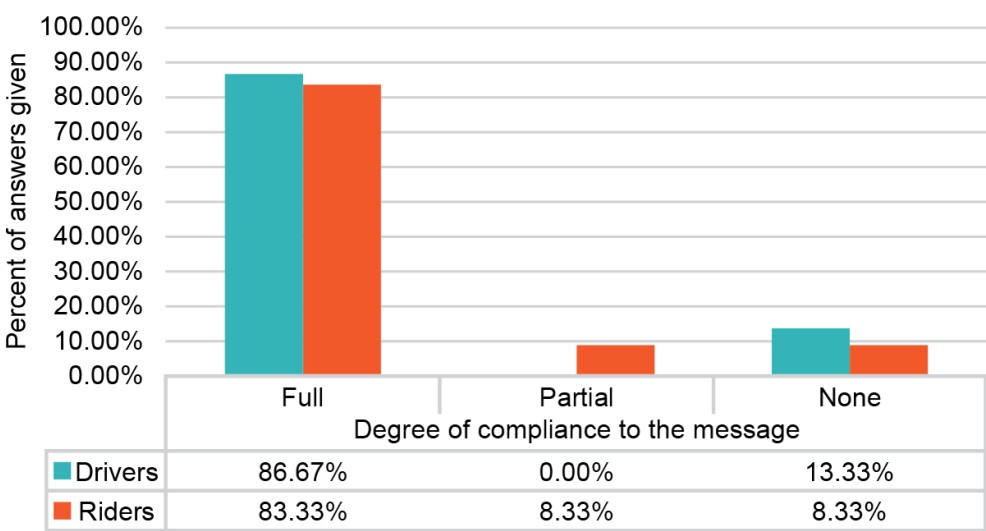

**Figure 16.** Degree of compliance for the scenario "VMS".

Regarding compliance, the sum of the results indicated that for the drivers, the success rates were close to 90% for all scenarios, while for the riders this number was close to 80%. Partial success was rare and ranged between 0% for some scenarios and up to 10% for others, while the full failure of the trial was constantly close to 10% with the riders showing slightly higher percentages by 2% compared to drivers.

The second dimension of user acceptance is derived from the analysis of the provided questionnaires. From these questionnaires, it was possible to draw conclusions about the profile of the users and their opinion regarding the experience with the system.

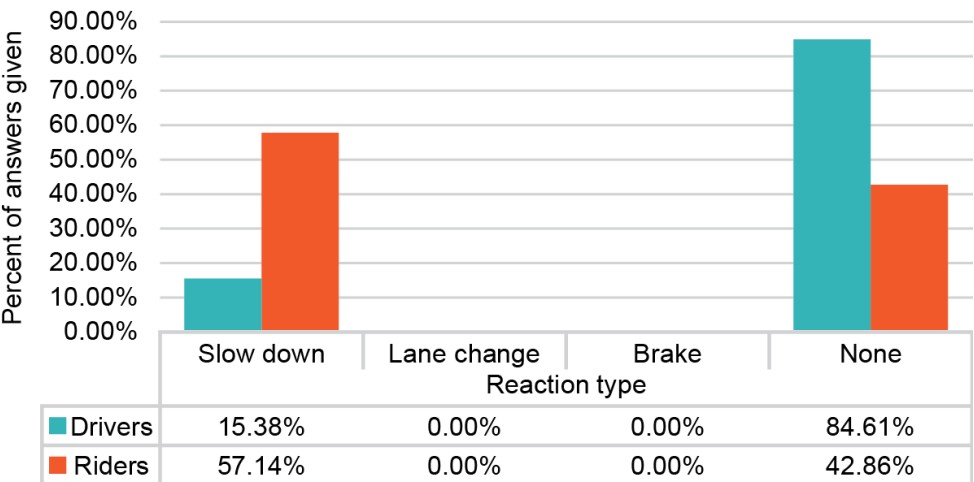

| | Slow down | Lane change | Brake | None |
|---|---|---|---|---|
| Drivers | 15.38% | 0.00% | 0.00% | 84.61% |
| Riders | 57.14% | 0.00% | 0.00% | 42.86% |

**Figure 17.** Recorded reactions for the scenario "VMS".

### 5.2.1. Results Derived from the Pre-Test Questionnaires

The gender representation amounted to 64.71% for men and 35.29% for women, however, the extracted results refer to all users and were not distinguished based on gender. A total of 33.33% of users have driving experience of up to ten (10) years, 40% have from 11–20 years and 26.67% have 21–30 years. Out of these, it is noteworthy that only 18.75% drive a car that is less than five (5) years old in their daily commutes, with the majority driving older generation vehicles. Most of these drivers (88.24%) indicate that they use their vehicles on a daily basis. Exceeding the speed limit and illegal parking are the most common traffic violations among users, followed by passing through a red light (answered by almost 30%), speeding, and driving under the influence of alcohol.

Users reported they have been involved in an average of 1.41 accidents per user in years of driving. Regarding the drivers' prior experience with the use of advanced driving assistant systems (ADAS) in their vehicles, the first place takes the use of navigation together with the parking assistance system, which seems to have been integrated into the daily life of most drivers. Blind spot detection and forward collision warning systems follow after with quite less frequency of use. It is noteworthy that applications/systems that belong to the category of road safety have a greater frequency of use, while other auxiliary systems, such as cruise control and parking assistance, appear last.

Similar to the drivers/riders, the road operators completed pre-test questionnaires, specific to their expertise. Of the respondents, a large percentage (66.7%) have no expertise in road condition monitoring and no answer was recorded for the use of software related to road maintenance and inspection. In the question about what kind of system they think would best suit the needs of their position, they stated (a) Inspection Management System and (b) Asset Management System. Half of them (50%) also responded positively to the question of whether their organization uses predictive analysis software or other decision support module for scheduling a maintenance plan. The operators consider the timely renewal of the transmitted information as an advantage of the existing infrastructure of variable message signs (VMS) but recognize the small number and sparse location as a big drawback. A small number also mentioned the inability to display images for quick and easy information as a disadvantage for the existing VMS.

### 5.2.2. Results Derived from the Post-Test Questionnaires

Driver/riders generally consider the system to be reliable (Figure 18) as the large majority tend to evaluate positive features, such as the sharp information received, but there is also a large percentage of neutral evaluation from a segment of the users, without indicating that this factor would be a deterrent for them to use the system. The vast majority of the drivers' intention is to use the system in the future if possible.

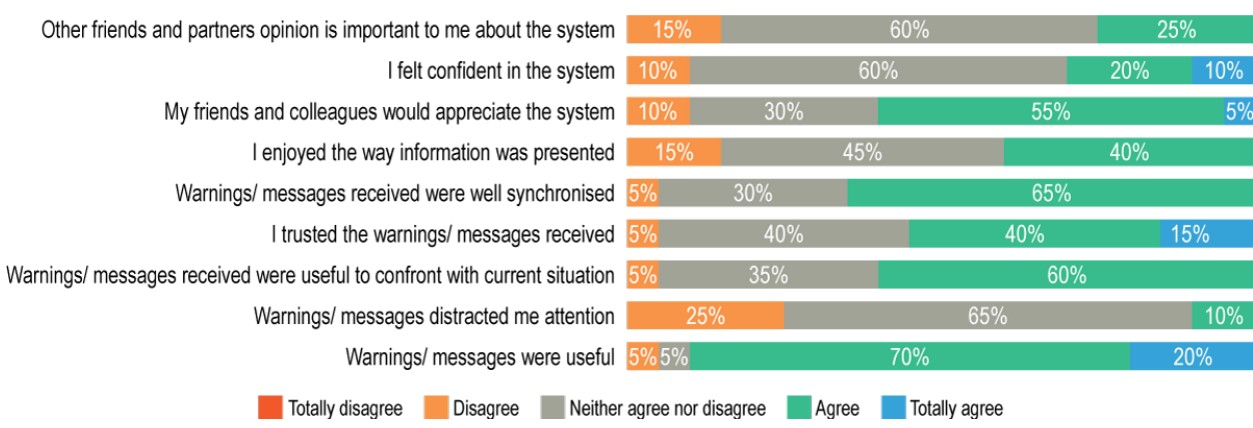

**Figure 18.** Usability evaluation of the application.

An important element for further development of this technological innovation is the evaluation of the workload (driving duty) that the system imposes on the driver. The most demanding actions and therefore with the biggest workload the display of information (eye distraction) and then the requirements placed on the drivers regarding the timely response, as presented in Figure 19.

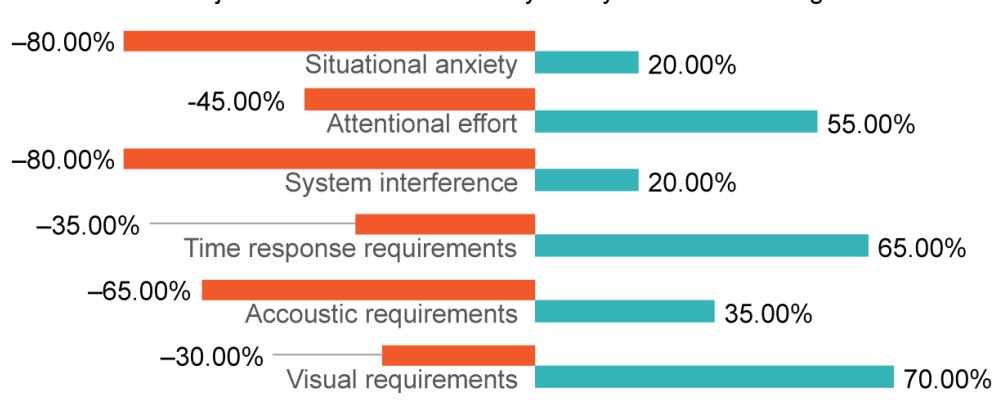

**Figure 19.** Subjective workload perceived by the users.

In the qualitative evaluation (free text), users singled out valid and timely information, especially on security issues, as advantages of the system; the beautiful design and the nice presentation of the information were noted as particularly positive elements. The absolute majority of the users indicated that they would like more audio warnings as well as more options for connectivity (e.g., via Bluetooth), while some users also mentioned that personalization in the sound would be most welcomed so that they can choose different audio for each case or warning. In addition, the users stated that they would like to have on display the estimated time to an event (e.g., collision time in the case of a crash) when this is for security reasons.

In total, 95% of users state that they expect the system to provide a positive impact on their daily mobility, but the largest amount (85%) is willing to pay a price of up to €20 to get it and 15% of users intend to pay more, in the range between 20–50 €. Regarding the importance of the expected effects, the users singled out the enhancement of the penetration of C-ITS systems following other impacts, without, however, giving particular importance to any, as the score is relatively close. The expected impact of the system is presented in Figure 20.

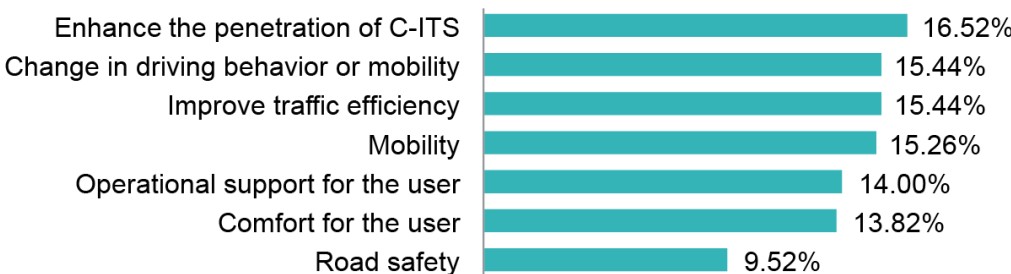

**Figure 20.** Expected impacts of the system.

From the evaluation of the user acceptance from the side of the road operators, the results are very positive for the particular features that strengthen the acceptance of the system. The respective reliability results for the operators show a big confidence of the users as much for the functionality of the system as well as the provided information. Regarding the market penetration, all road operators would consider themselves as future users of the system and like to use it, while the majority (60%) of them intend to pay for this use. The great majority of these (40% with intensity of nine (9) out of ten (10), 40% with ten (10) out of ten (10) and 20% with seven (7) out of ten (10)) would recommend the system to a colleague. Among the advantages of the system, the majority mention the timely and immediate provision of information about the condition of the road coating, believing that the system can work positively as a complementary application to existing maintenance systems. Among the disadvantages, there is a single reference for the provision of notifications through acoustic messages, while no proposal for a change in the existing form of the application was recorded. In terms of the legal/regulatory/operational barrier, no user stated a barrier to the operation of the system and none foresaw any conflict with other existing or emerging technologies.

## 6. Discussion

Based on the pilot tests, the applications produced a smoother driver reaction, giving the driver more time (than in real-life) to correct maneuvers and thus reduce the level of risk. In other words, the system allows drivers to be warned and to understand and implement necessary actions in advance. This was particularly evident with the roadworks application, where the driver received information about a closed lane ahead in a much shorter time than they themselves would have seen in real-life, resulting in a very smooth, gentle deceleration and change in the lane. Smooth maneuvers and driving behaviors have beneficial effects on traffic flow and reduce the risk of head-on collisions. In general, early notification of the upcoming conditions on the road ahead provides the potential to apply more appropriate maneuvers with a substantial positive impact on avoiding collisions.

Avoiding collisions on the road has, in turn, a positive impact on the road infrastructure operators, as incidents on the road are reduced and traffic is kept smooth. The ability of the road operators to communicate directly with the drivers in the area where they will encounter imminent danger plays an important role in this as not only do the drivers become recipients of the information, they can also transfer critical information (through the vehicle to infrastructure communication) to the traffic management center.

The level of user acceptance, as recorded by the pre and post-questionnaires is very high with even more expectations about future use of the system in their daily commute. Possibly, a reason for such a degree of acceptance, is the ability of the system (through the selected communication channels) to upgrade the existing fleet of vehicles, producing immediate results without this cost being transferred to the users. At the same time, all stakeholders can enjoy the advantages, as in addition to the drivers, the infrastructure operators foresee that if the system is used successfully as a road pavement health monitoring

assessment system and better schedule of the maintenance plan, the profits in life cycle maintenance savings could be expected in the order of 30%.

Because of the system's good user acceptance, the authors intend to further expand the system in order to be able to cover more real use case scenarios and expand its service gamut by providing even more meaningful warnings to drivers. Energy efficiency and autonomy of the system are also key parameters that the authors intend to invest in during the system's further development phase so as to reduce the required maintenance cost. This is ultimately going to lead to a system that will be able to guarantee road safety with minimal purchase, installation and maintenance costs. Furthermore, the developed system, being an IoV device, needs to be able to provide future-proof services by incorporating better connectivity options through next-generation networks, such as 5G or 6G networks, while consuming less power [35]. The authors intend to explore various connectivity options for the next iteration of the system in order to achieve better connectivity and extended autonomy.

## 7. Conclusions

In the present work, an integrated IoV system is presented for enhancing road safety by monitoring various traffic and road behavior parameters and by sending specialized warning messages in case the traffic conditions or a certain behavior of the driver is prone to lead to an accident. The system comprises dedicated hardware and software developed for the project and employs versatile technologies and communication protocols that make the system scalable for use in large road and highway/motorway infrastructures, while it can be used by virtually all types of vehicles, despite their age and connectivity capabilities. The system's technical performance and user acceptance were evaluated during integration and field tests that were carried out in an actual motorway environment with two different types of vehicles. The system's performance in all aspects was very good and showed its potential in accomplishing its goal. The authors intend to further improve the system's usability by providing more warnings for more real use cases, expand the system's connectivity features by incorporating next generation connectivity options, improve the system's autonomy and reduce its installation and maintenance cost.

**Author Contributions:** Conceptualization, A.S., F.G., C.S., M.G. and N.K.; Validation, N.K.; Formal analysis, F.G., C.S., M.G..; Data curation, A.S., F.G., C.S., M.G.; Writing – original draft, A.S., F.G; Writing – review & editing, C.S., M.G.; Funding acquisition, A.S. All authors have read and agreed to the published version of the manuscript.

**Funding:** This research received no funding.

**Acknowledgments:** This research has been co-financed by the European Union and Greek national funds through the Operational Program Competitiveness, Entrepreneurship, and Innovation, under the call RESEARCH-CREATE-INNOVATE (project code: T1EDK-03081).

**Conflicts of Interest:** The authors declare no conflict of interest.

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
