# Peer review of "Integration and Field Evaluation of an IoV System for Enhancing Road Safety"

_applsci, doi:10.3390/app122312262_

Round 1

Reviewer 1 Report

The paper in question deals with a very topical issue, consisting in the use of telematics in vehicles in order to improve traffic safety and reduce accidents. A system called ODOS2020 is proposed, based on innovative components and on the use of the cloud. Functionality tests were conducted on a recently built stretch of motorway, with 370 transits, followed by questionnaires. The results were positive in terms of technical performance of the system, with limited deviations in terms of speed. Acceptance of the prototype by users, who have provided ideas for improvement, seems to be good.

Overall I consider the article worthy of publication, but after improvement of figure 1, figure 2 and figure 8 (they cannot be read well), and a better presentation of the data in table 3 and table 4 (there are no vertical lines separating car and motorcycle , as well as between error% and speed).

Reviewer 2 Report

In this paper, the authors evaluated the ODOS2020 IoV system to improve road safety. Firstly, they introduce an overview of the architecture and features of the ODOS2020, then, they design tests to evaluate system performance. In the reviewers' opinion, this work is topical and attractive, however, it should be major revised to enhance the quality, as follows:

1) Obviously, IoVs are integrated systems of VANETs, the Internet, and smart computing methods. However, I don't find your mentions of the VANETs concept and its characteristics such as pattern mobility, reply RSU, high movement speed, etc., and its challenging problems. The authors should indicate these issues in Introduction Section, refer in (Communication solutions for vehicle ad-hoc network in smart cities environment: a comprehensive survey). 

2) All Figures should be enhanced with a minimal resolution at 300 dpi. Figures 1, 3, 5, 8, and 15-20 are difficult for readers. Figure 2 is missing.

3) There are many typos/writing/format. The authors should double-check throughout the paper, for example:

- Lines 265-274, reformat Equations;

- "2 categories" rewritten "two categories"

4) In Section 6 (Discussion), the authors should indicate some challenges and future study directions. Moreover, you also should mention the relationship between IoV applications in the future Internet context such as 5G, and 6G. This will highlight this work's contributions, refer in (Wireless Communication Technologies for IoT in 5G: Vision, Applications, and Challenges).

Round 2

Reviewer 2 Report

This version is improved significantly compared to the original manuscript. I think it is eligible for publishing.